# Skin-to-Skin Contact: Crucial for Improving Behavior, Immunity, and Redox State after Short Cohabitation of Chronologically Old Mice and Prematurely Aging Mice with Adult Mice

**DOI:** 10.3390/ijms24054680

**Published:** 2023-02-28

**Authors:** Estefanía Díaz-Del Cerro, Judith Félix, Mónica De la Fuente

**Affiliations:** 1Department of Genetics, Physiology, and Microbiology (Unity of Animal Physiology), Faculty of Biology, Complutense University of Madrid (UCM), 28040 Madrid, Spain; 2Institute of Investigation 12 de Octubre (i+12), 28041 Madrid, Spain

**Keywords:** skin-to-skin contact, aging, mice, behavior, immune system

## Abstract

Aging is characterized by a deterioration of the homeostatic systems, namely the nervous and immune systems. The rate of aging can be modified by lifestyle factors such as social interactions. Recently, improvements in behavior, immune function, and oxidative state were observed in adult prematurely aging mice (PAM) and chronologically old mice after cohabitation with exceptional non-PAM (E-NPAM) and adult mice, respectively, for 2 months. However, the cause of this positive effect is not known. The objective of the present work was to study whether skin-to-skin contact promotes these improvements both in chronologically old mice and in adult PAM. Old and adult CD1 female mice were used as well as adult PAM and E-NPAM. After cohabitation for 15 min/day for 2 months (two old mice or PAM with five adult mice or E-NPAM, respectively, with both non- and skin-to-skin contact), several behavioral tests were performed and functions and oxidative stress parameters in peritoneal leukocytes were analyzed. This social interaction improved behavioral responses, immune functions, redox state, and longevity, but only if the animals had skin-to-skin contact. Physical contact seems to be crucial to experiencing the positive effects of social interaction.

## 1. Introduction

The homeostatic systems such as the nervous and immune systems suffer deterioration with aging. Consequently, there is an age-related loss of homeostasis and health [1,2]. For this reason, aging is associated with a higher risk of morbidity and mortality. This deterioration is due to the establishment of chronic oxidative-inflammatory stress; that is, there is an imbalance between the oxidizing and inflammatory compounds and their defenses in favor of the former [2,3].

Aging is also a highly heterogeneous process, which explains how individuals of the same chronological age can show a different rate of aging, and one of the factors that influences this is the capacity for stress response [1,4,5]. Thus, several years ago, we proposed a model of natural premature aging in mice based on the chronic hyperreactivity that they present in novel situations, such as when facing a T-maze. Adult mice that remained frozen and took a long time to cross the intersection of the T presented premature immunosenescence, as well as behavioral, oxidative, and hormonal alterations similar to those found in chronologically old animals. All this was reflected in a lower life expectancy with respect to their peers of the same age and sex who presented a very good response in the maze. The first group was called “Prematurely Aging Mice” (PAM), and the second group was called “Exceptional Non-Prematurely Aging Mice” (E-NPAM) [6,7,8,9,10,11,12].

Since the elderly population is growing exponentially in developed societies, which is accompanied by an increase in the appearance of age-related diseases, the field of research involving strategies capable of slowing down the aging process and thus achieving healthy longevity is increasing actively. In this context, it has been seen that the social environment in which one lives influences the health of the individual [13], both positively and negatively. In fact, in social species such as humans and mice, loneliness or isolation, as well as cohabitation with sick individuals [2,14,15,16], are situations that negatively affect behavioral and immune responses. However, the social environment may also cause beneficial effects. In this line, Garrido et al. found that after continuous cohabitation for two months of chronologically old mice with adults, or PAM with E-NPAM, the former experienced an improvement in both behavioral and immune function and, consequently, greater longevity [17,18]. However, these studies revealed that this cohabitation implied a deterioration in the homeostatic systems in both adult mice and in E-NPAM, although their longevity was not affected in any of the cases [17,18,19]. However, when the time of cohabitation is shorter, 15 min/day for 2 months, not only are these behavioral and immune improvements still observed in chronologically old mice and in PAM but also in adults, who achieve a longer lifespan [20,21]. Nevertheless, the physiological mechanisms underlying these effects are not known.

Although visual, olfactory, and auditory mechanisms of communication between mice could be involved in the effects observed after these social cohabitations, it is known that the sense of touch is of great importance in these kinds of interactions [22]. In fact, touch is the main form of interindividual communication in many animal species. In addition, evolutionarily, it precedes oral communication [23], suggesting that touch may play an important role in social interaction. Due to this, and given the ease of the experimental approach, in the present work, we proposed to study the influence of skin-to-skin contact on behavior, immunity, and redox state in chronologically old mice and PAM after cohabiting for 15 min/day for two months with chronologically adult mice and E-NPAM, respectively.

## 2. Results

### 2.1. Physical Contact Mediates Improvement of Behavioral Responses in Old Mice/PAM after Social Interaction with Adults/E-NPAM

The results of the behavioral tests performed after the short social cohabitation between old and adult mice as well as PAM and E-NPAM are shown in Figure 1 and Table 1. With respect to the sensorimotor tests of the wood-rod and tightrope tests (Figure 1), old mice and PAM exhibited deteriorated coordination, balance, traction, and exploratory abilities in comparison with adult mice and E-NPAM, respectively. However, old mice that had skin-to-skin contact with adult mice improved these abilities since they took less time to complete the wood-rod test (Figure 1(A1), *p* < 0.05), and the percentage of animals with optimal traction was higher (Figure 1(B1), *p* < 0.05) than in the control groups (C and NSC groups). A similar effect can be observed in PAM that had skin-to-skin contact with E-NPAM (Figure 1(A2,B2), *p* < 0.05). In addition, old mice in the skin-to-skin contact (SC) group presented a lower latency when leaving the starting segment in the wood-rod test and a higher total number of rearings performed in the elevated plus maze (Table 1, *p* < 0.05) than old mice in the control (C) group but not compared to old mice in the NSC group. Moreover, PAM SC presented a higher latency when falling in the tightrope test (*p* < 0.05) than PAMC and PAM NSC (Table 1, *p* < 0.05). Additionally, adult mice of the SC group presented higher total and external locomotion (*p* < 0.05) in the hole-board test than the C group but not higher than the NCS group. SC E-NPAM showed a higher latency when falling in the tightrope test (*p* < 0.01) and total locomotion in the hole-board test (*p* < 0.05) than the C and NSC E-NPAM groups (Table 1).

In addition, in the results obtained with respect to anxiety levels (Figure 1C), we can see that the old mice and PAM of the control groups (C) exhibited more anxiety-like behavior than adult mice and E-NPAM, respectively, because they spent less time in the open arms in the elevated plus maze (*p* < 0.05). However, old mice and PAM, after physically interacting with adults and E-NPAM (SC groups), respectively, spent a higher percentage of time in these open arms (*p* < 0.05) than the C and NSC groups, as well as less time in closed arms (*p* < 0.01 and *p* < 0.05, respectively (Table 1)).

### 2.2. Physical Contact Mediates Improvement in the Immunity of Old Mice/PAM after Social Interaction with Adults/E-NPAM

The results for the immune functions are shown in Figure 2 and Table 2. With respect to these functions, old mice and PAM showed deteriorated chemotaxis index of macrophages and lymphocytes, phagocytosis natural killer activity, lymphoproliferation in response to mitogens (ConA and LPS), and basal proliferation (Figure 2 and Table 2) in comparison with adult mice and E-NPAM, respectively. However, the SC group constituting old mice had a greater chemotactic capacity of lymphocytes (Table 2, *p* < 0.05), natural killer activity (Figure 2(A1), *p* < 0.05), and lymphoproliferation in response to ConA (Figure 2(B1), *p* < 0.05) than the old control (C) group and the old mice that did not physically contact adults (NSC). The PAM after skin-to-skin contact with E-NPAM (SC group) also presented some improved immune functions. Thus, these presented higher phagocytic index (Table 2, *p* < 0.01), cytotoxic NK activity (Figure 2(A2), *p* < 0.01), and lymphoproliferation values in response to ConA (Figure 2(B2), *p* < 0.05) values than those in the PAMC group. No difference was observed in the chemotaxis capacity of macrophage and in LPS-stimulated lymphoproliferation.

As for adults who physically interacted with old mice (SC group), they showed no change in their immune functions in comparison with the C and NSC groups. However, E-NPAM that had skin-to-skin contact with PAM (SC group) had lower levels of NK activity (Figure 2(A2), *p* < 0.001), chemotaxis of lymphocytes (*p* < 0.05) (Table 1), and lymphoproliferation in response to ConA (Figure 2(B2), *p* < 0.01) in comparison with C E-NPAM.

Basal lymphoproliferation (Figure 2C), a parameter of inflammation, was lower in old mice (*p* < 0.05) and PAM (*p* < 0.01) than in skin-to-skin contact adults and E-NPAM, respectively, with respect to their control groups (C and NSC groups, respectively). No change was observed in the adult and E-NPAM experimental groups.

### 2.3. Physical Contact Mediates Improvement in the Redox Balance of Old Mice/PAM after Social Interaction with Adults/E-NPAM

The effects observed on the oxidative state of the peritoneal leukocytes of these animals are shown in Figure 3 and Table 3. With respect to these parameters, old mice and PAM showed lower levels of antioxidant defenses (GPx activity and GSH concentration) and higher amounts of oxidant compounds (GSSG, TBARS) than adult mice and E-NPAM, respectively. It can be observed that the SC groups constituting old mice and PAM had lower levels of GSSG (Figure 3A, *p* < 0.05), GSSG/GSH ratio (Figure 3B, *p* < 0.05), and TBARS (Figure 3C, *p* < 0.05) in their peritoneal leukocytes compared to their control groups (C and NSC groups, respectively). The SC group constituting old mice also showed higher GPx activity (Table 3, *p* < 0.01) and higher concentrations of GSH (Table 3, *p* < 0.05) than old mice from the C and NSC groups. No changes were observed in the adult and E-NPAM experimental groups. Moreover, no difference was observed in the GR activity in any case (Table 3).

Finally, the results of this cohabitation on mean longevity and cumulative survival are shown in Table 4. The PAM C group showed lower mean longevity and cumulative survival than the E-NPAM C group (*p* < 0.05). Old mice and adult mice that cohabited with contact (SC groups) presented higher cumulative survival than the corresponding controls (*p* < 0.05). In the case of old animals and PAM, these values in the SC groups were also higher than the corresponding NSC groups (*p* < 0.05). No effect on longevity was observed in the E-NPAM groups.

## 3. Discussion

In this study, it has been shown that both chronologically old mice and prematurely aged adults improve various behavioral responses, as well as their immunity and redox state, after interacting socially and having skin-to-skin contact with adults or E-NPAM, respectively. In this context, it is known that elderly individuals present a deterioration in their social communication [24], and one could think that the same would be the case for PAM. Therefore, interacting with adult individuals or E-NPAM can make them develop mechanisms that improve their social communication through various channels, such as auditory, olfactory, visual, tactile, or all of them. In this line, this study provides the first evidence for the effect of physical contact in this type of positive social environment strategy, thus demonstrating the crucial role that such contact has for animals to improve the parameters analyzed.

Several studies have shown that physical contact, in addition to transmitting information, emotions, or social support [23], is also capable of affecting people’s health, producing positive biological stimulation [22]. The skin is considered a neuroimmunoendocrine organ since it contains epidermal cells and non-epidermal cells such as immune or nervous cells, which influence the regulation of the homeostatic system [25]. Thus, the skin is associated with the peripheral sensory nervous system, the autonomic nervous system, and the central nervous system [25], with the C fibers being responsible for transmitting social sensory information to the brain [22]. Therefore, tactile stimulation is able to modulate the effects of cohabitation on the behavioral parameters analyzed. In fact, the results obtained in the present work using skin-to-skin contact (SC) groups (old mice and PAM) show an improvement in their sensorimotor skills (i.e., motor coordination, balance, and neuromuscular vigor) with respect to old mice and PAM that cohabited with adults or E-NPAM without skin-to-skin contact (NSC groups). It has also been observed that, in non-human mammals, tactile stimulation between conspecifics has analgesic effects [23]. Moreover, physical contact has been suggested as a stress buffer, playing a critical regulatory role in the body’s responses, including cortisol responses, which ultimately promote social connection [26]. This coincides with the results obtained in this study as old mice and PAM that had skin-to-skin contact with adults and E-NPAM, respectively, had lower levels of anxiety compared to the old mice and PAM that cohabited without physical contact with adults and E-NPAM, respectively.

As a neuroimmunoendocrine organ, the tactile stimuli perceived by the skin can influence immunity. In fact, the results obtained in the present work showed that the old mice and PAM that maintained physical contact with the adults and E-NPAM (SC groups), respectively, presented several improved relevant immune functions (chemotactic capacity, phagocytosis, natural killer activity, and ConA-stimulated lymphoproliferation) with respect to the old mice and PAM without physical interaction with adults and E-NPAM, respectively (NSC groups).

Since the age-related deterioration in immunity underlies oxidative stress [2,27], in the immunosenescence that presents in old mice and adult PAM, which has been confirmed in the animals in the present work, we should expect the improvement in the immune system by the skin contact observed in the old mice and PAM after cohabitating with skin-to-skin contact with regular adult mice and E-NPAM, respectively, would to be associated with a decrease in the oxidative stress of these individuals. The results obtained in this study support this. Thus, the SC groups of old mice and PAM showed lower values of oxidative compounds (concentration of oxidized glutathione and of lipid peroxidation) and higher values of antioxidant glutathione peroxidase activity (in the case of SC old mice), as well as a lower GSSG/GSH ratio. Since this ratio is an indicator of oxidative stress [28], it seems evident that the old mice and PAM that cohabited with skin-to-skin contact (SC groups) have a lower oxidation state than those that did not have skin-to-skin contact.

The immune functions and oxidative parameters analyzed in the present work have been proposed as markers for the rate of aging and as predictors of life span [10,11]. The improvement in these functions in PAM, a group with a significantly shorter lifespan than E-NPAMC, as several studies have shown [7,8,9,10,11,12], after the skin-to-skin interaction with E-NPAM, is reflected in an increase in lifespan with respect to the PAM of the control group and those who cohabited with E-NPAM without physical contact. This improvement in longevity was also observed in old and adult SC groups, in agreement with previous results [17,20], but it disappeared if the cohabitation was without contact (NSC groups). The E-NPAM groups did not show differences in longevity. Thus, in the E-NPAM group, this physical contact is not able to extend longevity, which could be a consequence of these animals being exceptional in many respects, including their life span [7,10,11,29].

Despite all the results obtained, the present study presents limitations. One is that there are no data on the type and amount of direct physical interactions between individuals during the time of social interaction. Therefore, in future studies, it would be interesting to relate the different types of physical contact and contact time to the degree of improvement in the parameters studied during this type of cohabitation. Another limitation presented by the study is that other mechanisms that are also involved in the beneficial effects obtained in this type of social interaction may be having a summative effect with physical contact. In fact, some of these mechanisms may be visual, olfactory, and auditory perception as they change with advancing age or health status [24,30,31,32]. Another possible mechanism could be the ingestion of fecal balls, which would lead to the alteration of the microbiota [33]. However, we see this mechanism as being more possible if the cohabitation was continuous, but with interaction rates of 15 min a day, the mice hardly defecated, and we did not observe coprophagous behaviors.

## 4. Materials and Methods

### 4.1. Animals

Ex-reproductive female mice (Mus musculus) of the ICR-CD1 strain (Janvier, France) were used. All the animals were housed, 5 individuals per cage, and placed in the Animal Facility at the Faculty of Biology (Complutense University of Madrid, UCM, Madrid, Spain) at 22 ± 2 °C, 60% humidity. They were maintained at a 12/12 h reversed light/dark cycle (lights on at 20:00 h) to avoid circadian interference. Mice were checked daily. Water and standard pellets (Panlab, Spain) were available ad libitum. The diet used followed the recommendations of the American Institute of Nutrition for Laboratory Animals (A04 diet from Panlab S.L., Barcelona, Spain). The protocol was approved by the Experimental Animal Committee of the Complutense University of Madrid and the Community of Madrid (PROEX. 224.0/21). All mice were treated according to the guidelines of Royal Decree 118/2021, 23 February 2021 (BOE No. 47), on the protection of animals for experimentation and other scientific purposes. We used the ICR-CD1 strain due to its higher genetic heterogeneity than inbred strains, facilitating the extrapolation of the results to humans [34], and using females, which are more suitable for co-housing studies due to the fact that they do not show the aggressive and dominant behavior of males, which could interfere with the results obtained in the parameters studied [17]. In addition, the females used were also ex-reproductive females due to their better extrapolation to humans, since it simulates a situation similar to human aging. Due to the use of females, after the cohabitation and before performing the tests, we confirmed that all adult mice were in the diestrus phase by vaginal smear since the phase of the estrous cycle they are in can affect the parameters studied in this study. With respect to old females, the cyclicity of their estrous cycle ceases (maintaining the diestrus phase), so the hormones do not affect the parameters studied in the present work [35].

### 4.2. Classification of Prematurely Aging Mice (PAM) Using the T-Maze

A total of 100 mice were subjected to the T-maze to be classified as PAM or E-NPAM. The T-maze consists of two wooden arms, one shorter (27 cm) that the other (64 cm), and is made from black methacrylate. The test was performed by placing the mouse inside the end of the short arm of the maze, facing the wall, and timing how long it took the animal to cross with its two hind legs the intersection of the arms. This test was performed once a week for 4 consecutive weeks. Mice that took more than 10 s to cross the intersection all 4 times were considered “Prematurely Aging Mice” (PAM), while those that took less than 10 s during the 4 consecutive weeks were classified as “Exceptional Non-PAM” (E-NPAM) [29].

### 4.3. Experimental Design

On the one hand, 12 old female mice (72 ± 4 weeks) and 21 adult female mice (32 ± 4 weeks) were used, which were divided into the following experimental groups: a control (C) group of old (n = 6) and adult mice (n = 7), a group of old mice that did not have skin-to-skin contact with adults (NSC, n = 6), a group of old mice that had skin-to-skin contact with adults (SC, n = 6), a group of adult mice that did not have skin-to-skin contact with old mice (NSC, n = 15), and a group of adult mice that had skin-to-skin contact with old mice (SC, n = 15).

On the other hand, after the PAM classification (mice from a different cohort than those in the previous study), we used 18 PAM and 37 E-NPAM, which were divided into the following groups: a control group (C) of PAM (n = 6) and E-NPAM (n = 7), a group of PAM that did not have skin-to-skin contact with E-NPAM (NSC, n = 6), a group of PAM that had skin-to-skin contact with E-NPAM (SC, n = 6), a group of E-NPAM that did not have skin-to-skin contact with PAM (NSC, n = 15), and a group of E-NPAM that had skin-to-skin contact with PAM (SC, n = 15).

In both experimental designs, the SC and NSC groups cohabited 15 min/day for two months at a proportion (per cage) of 2 old mice/PAM:5 adult mice/E-NPAM, respectively. The NSC groups cohabited in cages divided into two compartments separated by a transparent methacrylate wall. One was larger (to contain 5 animals) and another smaller (in which 2 animals were placed), and the SC groups inhabited standard cages, allowing them to have physical contact.

### 4.4. Behavioral Tests

#### 4.4.1. Wood-Rod Test

The wood-rod test was performed to evaluate motor coordination and equilibrium following the protocol previously described [36]. The test consists of a suspended wooden board 2.9 cm wide and 80 cm long. The test begins by placing the animal in the center and lasts 60 s. Balance ability was evaluated using the time it took to complete the test (in seconds), while motor coordination was assessed using the latency time (in seconds) when leaving the starting segment.

#### 4.4.2. Tightrope Test

The tightrope test was used to evaluate motor coordination, neuromuscular vigor, and traction [7,36]. It consisted of a suspended 60 cm long hemp rope divided into segments of 10 cm. The test begins with the placement of the animal in the center of the rope, suspended only with the help of the front legs. The duration of the test is 60 s. Neuromuscular vigor was determined using the percentage of mice that fell and the latency of the fall (in seconds). Traction was analyzed by observing the different parts of the body (forelimbs, hind limbs, and tail) that mice used to remain hanging and with the percentage of mice displaying the optimal traction capacity (employing the three parts of the body).

#### 4.4.3. Elevated-Plus Maze

Anxiety levels were determined using the elevated-plus maze test [37], which has two open and two closed arms. This test is based on mice’s aversion to open spaces and heights. The test lasts 5 min. For the assessment, the animal is placed on the central platform facing one of the closed arms. The animal is considered to enter one of the arms when it passes through it with all four legs. The parameters recorded were time in closed arms, time in the central platform, percentage of time in open arms, total number of head-dipping (number of times the animal introduces the muzzle or head in an arm to explore it), and rearing positions (vertical exploration) performed.

#### 4.4.4. Hole-Board Test

The hole-board test is a quadrangular apparatus divided into 36 squares divided into three zones (external, intermediate, and internal). In addition, in the central area, there are 4 equidistant holes, drawing a central area formed by four squares. The duration of the test is 5 min [36]. The parameters recorded for “non-goal-directed behavior” were external (number of squares that run in the external zone) and total (total number of squares crossed) locomotion. As a parameter of goal-directed behavior, the number of head-dippings performed (the number of times the animal introduces the muzzle or head inside a hole to explore it) was measured. Finally, as a parameter of vertical exploration, the number of rearing positions was measured.

### 4.5. Extraction of Peritoneal Leukocytes

The extraction of peritoneal leukocytes was performed following a previously described method [7] by injecting 3 mL of sterile Hank’s solution at 37 °C into the peritoneal cavity of the animal and its subsequent extraction by abdominal massage. Subsequently, the concentration of macrophages and lymphocytes, identified by their different morphologies, was counted using a Neubauer chamber, and cell viability was determined using the test of the vital dye blue trypan. Only cell suspensions whose viability was greater than 95% were used.

### 4.6. Evaluation of Immune Functions

#### 4.6.1. Phagocytic Activity

The phagocytic capacity of peritoneal cells was studied following a previously described procedure [10] based on the ability of macrophages to ingest inert particles (latex balls), which reflects their phagocytic function in vivo. The phagocytic index (the number of latex particles ingested per 100 macrophages) and the phagocytic efficacy (the percentage of macrophages capable of ingesting at least one latex particle) were determined.

#### 4.6.2. Chemotaxis Capacity

The chemotactic capacity of peritoneal cells was assessed following a modification of the Boyden method [10]. This technique is based on the ability of immune cells to move towards an infectious focus mimicked by a chemoattractant, formylated peptide (fMet-Leu-Phe) of Escherichia coli placed in the lower compartment of the Boyden chamber. The parameter evaluated was the chemotaxis index (C.I) of each cell type (the number of cells counted on the lower face of the filter that separates the two compartments of the chamber).

#### 4.6.3. Natural killer Cytotoxicity

Cytotoxicity was assessed using a previously described method [10] based on detecting the lysis of target cells (YAC-1 cell line of murine lymphoma) induced by peritoneal leukocytes (effector cells) through the determination of the enzyme lactate dehydrogenase using colorimetry. The results are expressed as a percentage (%) of lysis of the tumoral cells.

#### 4.6.4. Lymphoproliferation

The proliferative capacity of lymphocytes was performed following a previously described method [7]. The basal and stimulated capacity were assessed using the mitogens concanavalin A (ConA) and lipopolysaccharid (LPS) in cultures of 48 h after the incorporation of H3-thymidine. The results were expressed in counts per minute (c.p.m) for basal lymphoproliferation and in the percentage of stimulation (i.e., mitogen-stimulated lymphoproliferation divided by basal ×100) in cases of proliferative response stimulated in the presence of mitogens.

### 4.7. Oxidative Parameters

#### 4.7.1. Glutathione Peroxidase (GPx) Activity

The activity of this antioxidant enzyme was assessed using the protocol developed by Lawrence and Burk (1976) [38] but slightly modified [39], which is based on measuring the activity using cumene hydroperoxide (cumene-OOH) as a substrate. In this method, the oxidation rate of glutathione produced by cumene-OOH in a reaction catalyzed by glutathione peroxidase (GPx) is determined thanks to the decrease in absorbance at 340 nm due to the oxidation of NADPH in the presence of an excess of the enzyme glutathione reductase (GR). The results were expressed in milliunits (mU) of glutathione peroxidase activity per milligram of protein (mU GPx/mg protein).

#### 4.7.2. Glutathione Reductase (GR) Activity

Glutathione reductase (GR) activity was determined using the method described by Massey and Williams (1965) [40], which is based on the monitoring of NADPH oxidation used by the enzyme for the reduction of oxidized glutathione (GSSG). It was measured with absorbance and the results were expressed in milliunits (mU) of glutathione reductase activity per milligram of protein (mU GR/mg protein).

#### 4.7.3. Glutathione Concentration

For the assessment of the reduced state (GSH), the oxidized state (GSSG), and the redox ratio (GSSG/GSH), the method described by Hissin and Hilf (1976) [41], modified [11] and adapted to perform a plate analysis, was used. Fluorescence, using the fluorescent probe O-phtaldialdehyde (OPT), was measured at 350 nm excitation and 420 nm emission. The result is expressed in nanomol per milligram of protein (nmol/mg protein). The GSSG/GSH ratio was obtained by dividing the corresponding values of GSSG and GSH.

#### 4.7.4. Lipid Peroxidation: TBARS Concentration

The concentration of thiobarbituric acid reactive substances (TBARS) was evaluated using the commercial kit “Lipid Peroxidation (TBARS) Assay Kit” (BioVision, CA, USA) with some modifications [20]. The assay is based on the reaction of TBARS with 2 molecules of thiobarbituric acid (TBA), forming a pink compound with maximum absorption at 532 nm. The results were expressed in nmol TBARS/mg of protein.

### 4.8. Protein Concentration

This assessment was carried out with the biocinchroninic acid (BCA) test using the BCA kit, which is based on the reduction of Cu^2+^, generating Cu^+^ ions that bind to the BCA and form a colored compound that absorbs light at 562 nm [20].

### 4.9. Longevity Study

In order to evaluate the possible beneficial effects of cohabitation on the mean lifespan, mice were housed in the same conditions until their natural death, which was recorded.

### 4.10. Statistical Analysis

The statistical analyses of the results were carried out using Graphpad Prism 8.4.1. All results are expressed as the mean and standard deviation of the data obtained in each test. First, it was checked whether the data from each test followed a normal distribution using the Kolmogorov–Smirnov test, and the homogeneity of variances was tested using the Levene test. Subsequently, a comparison of the means was carried out using a non-parametric analysis (Mann–Whitney’s U test) and a parametric analysis (Student’s t-test) of independent variables to identify the differences between the study groups. In addition, the categorical variables were analyzed using a contingency test (Chi-square). In all cases, it was considered that there were no significant differences when a p-value greater than 0.05 was obtained, while a p-value less than 0.05 was considered significant, less than 0.01 was very significant, and less than 0.001 was highly significant.

## 5. Conclusions

From the above study, it can be concluded that skin-to-skin contact is crucial to achieving health improvements associated with this type of cohabitation in old, adult, and PAM animals, allowing them to experience healthy aging and extended longevity.

## Figures and Tables

**Figure 1 ijms-24-04680-f001:**
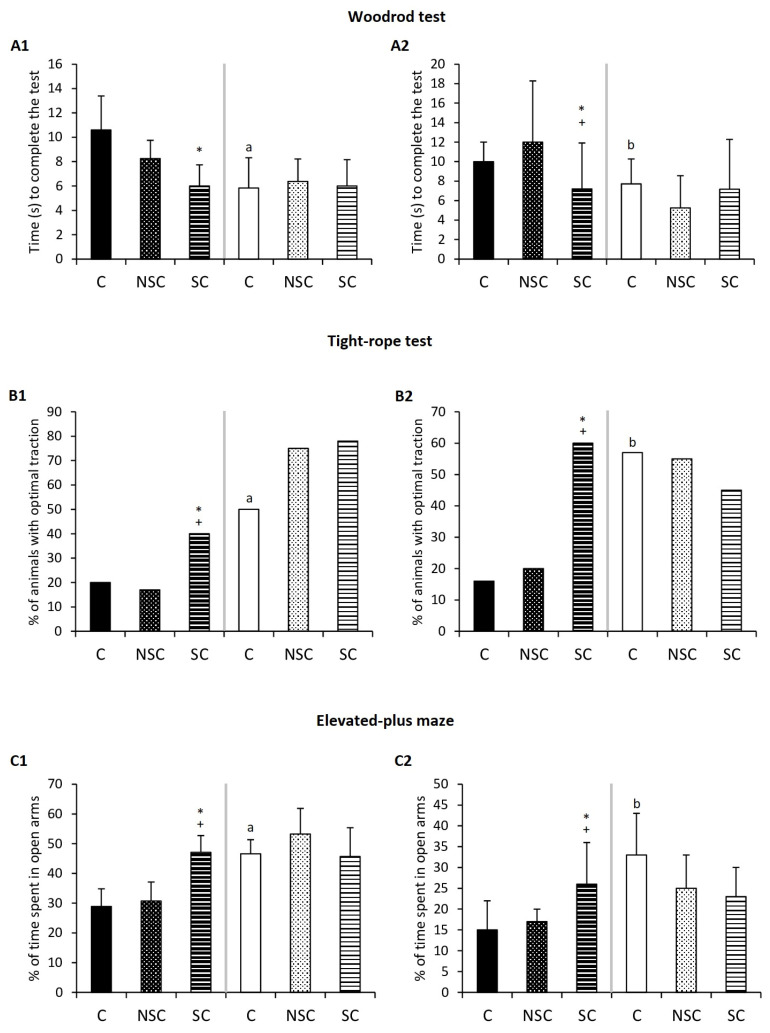
Behavioral parameters. (**A**) Time to complete the wood-rod test in seconds. (**B**) Percentage (%) of animals with optimal traction in the tight-rope test. (**C**) Percentage (%) of time spent in open arms in the elevated plus maze. The results corresponding to chronologically adult and old mice are shown in A1, B1, and C1, and to adult prematurely-aging mice (PAM) and non-prematurely-aging mice (NPAM) are shown in A2, B2, and C2. Each column shows the mean ± standard deviation of 6–15 values corresponding to the same number of animals. After a statistical analysis comparing the means with the Mann–Whitney U test for independent samples, significant differences were observed in the indicated experimental groups. *: *p* < 0.05 with respect to the values obtained in their corresponding control group. +: *p* < 0.05 with respect to the values obtained in the corresponding NSC group. a: *p* < 0.05 with respect to the values obtained in old mice C. b: *p* < 0.05 with respect to the values obtained in PAM C. C: control group (old mice/PAM: n = 6; adult mice/E-NPAM: n = 7). NSC: non-skin-to-skin contact (old mice/PAM: n = 6; adult mice/E-NPAM: n = 15). SC: skin-to-skin contact (old mice/PAM: n = 6; adult mice/E-NPAM: n = 15).

**Figure 2 ijms-24-04680-f002:**
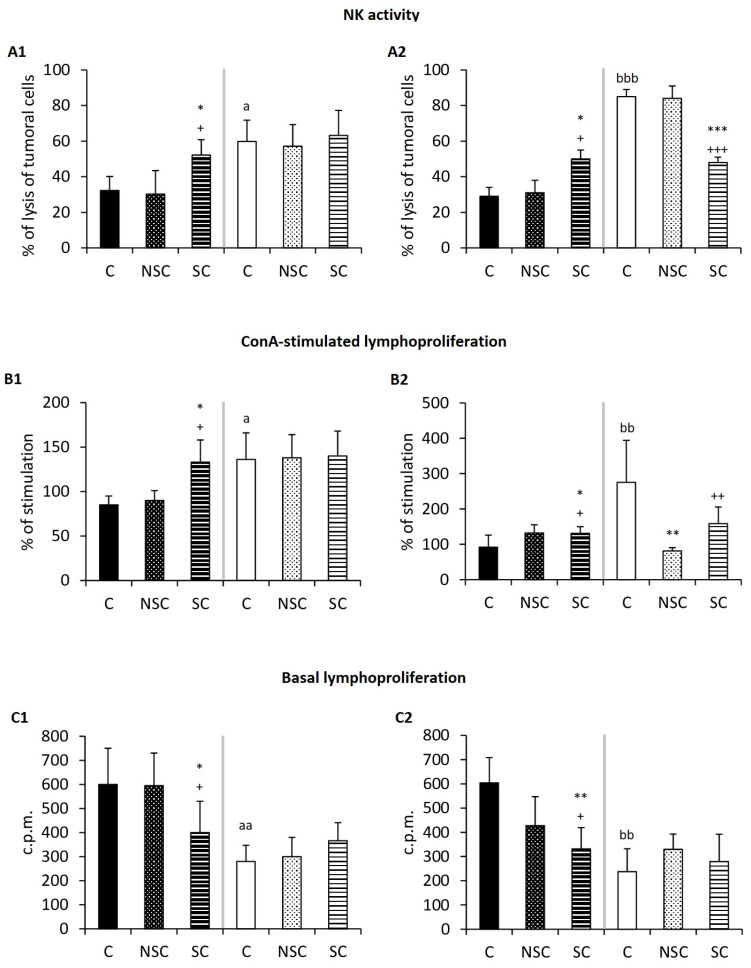
Changes in immune parameters. (**A**) Natural killer activity expressed as % of lysis of tumoral cells. (**B**) Percentage (%) of ConA-stimulated lymphoproliferation. (**C**) Basal lymphoproliferation in counts per minute (c.p.m.). The results corresponding to peritoneal leukocytes from chronologically adult and old mice are shown in A1, B1, and C1, and from adult prematurely-aging mice (PAM) and non-prematurely-aging mice (NPAM) are shown in A2, B2, and C2. Each column shows the mean ± standard deviation of 6–15 values corresponding to the same number of animals. After a statistical analysis comparing the means with the Student’s *t*-test for independent samples, significant differences were observed in the indicated experimental groups. *: *p* < 0.05, **: *p* < 0.01, ***: *p* ≤ 0.001 with respect to the values obtained in their corresponding control group. +: *p* < 0.05. ++: *p* < 0.01, +++: *p* ≤ 0.001 with respect to the values obtained in the corresponding NSC group. a: *p* < 0.05, aa: *p* < 0.01 with respect to the values obtained in old mice C. bb: *p* < 0.01, bbb: *p* < 0.001 with respect to the values obtained in PAM C. C: control group (old mice/PAM: n = 6. adult mice/E-NPAM: n = 7). NSC: non-skin-to-skin contact (old mice/PAM: n = 6; adult mice/E-NPAM: n = 15). SC: skin-to-skin contact (old mice/PAM: n = 6; adult mice/E-NPAM: n = 15). NK: natural killer. ConA: concanavalin. c.p.m.: counts per minute.

**Figure 3 ijms-24-04680-f003:**
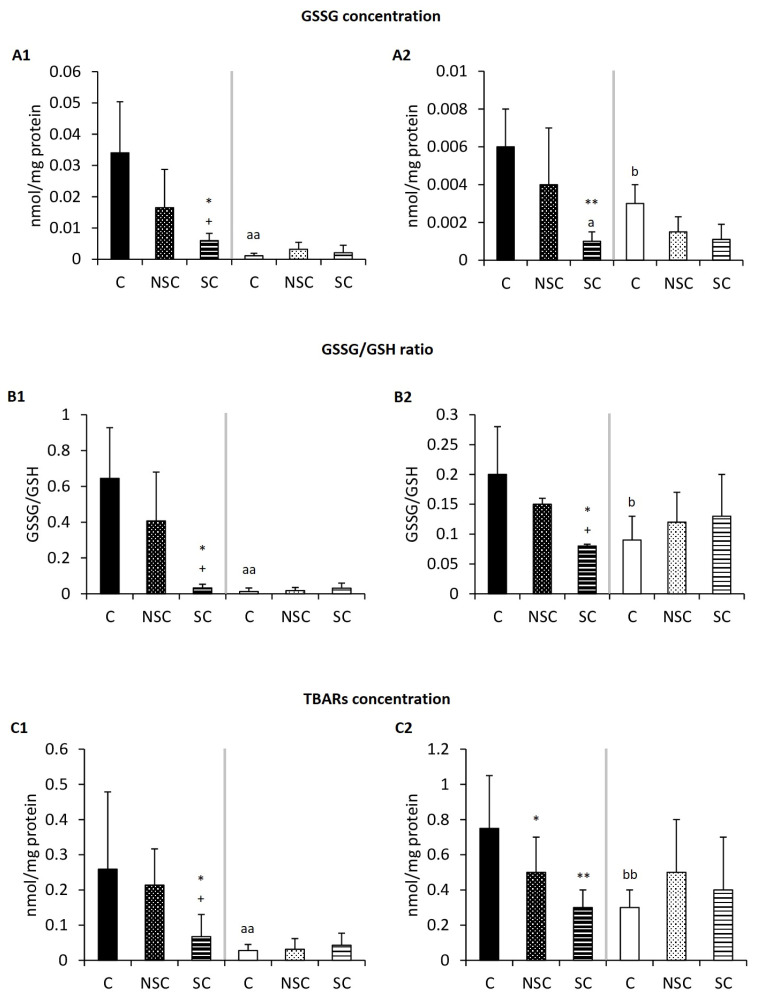
Changes in oxidative parameters in peritoneal leukocytes. (**A**) GSSG concentration in nmol/mg protein. (**B**) GSSG/GSH ratio. (**C**) TBARs concentration in nmol/mg protein. The results corresponding to peritoneal leukocytes from chronologically adult and old mice are shown in A1, B1, and C1, and from adult prematurely-aging mice (PAM) and non-prematurely-aging mice (NPAM) are shown in A2, B2, and C2. Each column shows mean ± standard deviation of 6–15 values corresponding to the same number of animals. After a statistical analysis comparing means with the Student’s t-test for independent samples, significant differences were observed in the indicated experimental groups *: *p* < 0.05, **: *p* < 0.01 with respect to the values obtained in their corresponding control group. +: *p* < 0.05 with respect to the values obtained in the corresponding NSC group. aa: *p* < 0.01 with respect to the values obtained in old mice C. b: *p* < 0.05, bb: *p* < 0.01 with respect to the values obtained in PAM C. C: control group (old mice/PAM: n = 6; adult mice/E-NPAM: n = 7). NSC: non-skin-to-skin contact (old mice/PAM: n = 6; adult mice/E-NPAM: n = 15). SC: skin-to-skin contact (old mice/PAM: n = 6; adult mice/E-NPAM: n = 15). GSSG: oxidized glutathione. GSH: reduced glutathione. TBARs: thiobarbituric acid reactive substances.

**Table 1 ijms-24-04680-t001:** Behavioral responses of old mice/PAM and adult mice/E-NPAM after cohabiting for 15 min/day for 2 months.

Behavioral Parameters	Old Mice			Adult Mice			PAM			E-NPAM		
	C	NSC	SC	C	NSC	SC	C	NSC	SC	C	NSC	SC
Wood-rod test												
Latency to leave the starting segment (s)	6 ± 1	4 ± 1	3 ± 0.4 *	3 ± 2 a	3 ± 2	3 ± 2	4 ± 2	4 ± 2	7 ± 4	4 ± 2	3 ± 2	3 ± 2
Tight-rope test												
Latency to fall (s)	14 ± 6	4 ± 2 *	14 ± 8 +	19 ± 10	26 ± 12	21 ± 15	10 ± 4	6 ± 3	30 ± 9 *	48 ± 10 b	19 ± 8 **++	4 ± 2 ***++
Animals that fell	5/6	5/6	3/6	2/7 a	2/10	3/10	5/6	6/6	3/6	4/7	4/6	4/7
Hole-board test												
Total locomotion	279 ± 24	353 ± 105	241 ± 27	250 ± 115	288 ± 64	369 ± 50 *	250 ± 36	310 ± 12 **	327 ± 29 **	281 ± 29	274 ± 26	324 ± 38 *+
External locomotion	158 ± 29	223 ± 93	147 ± 32	130 ± 73	165 ± 43	226 ± 45 *+	176 ± 27	193 ± 26	194 ± 17	146 ± 27	145 ± 20	153 ± 32
N° of rearings	17 ± 3	22 ± 13	22 ± 10	19 ± 7	21 ± 6	26 ±8	21 ± 6	15 ± 3	21 ± 7	22 ± 8	21 ± 8	20 ± 6
N° of head-dipping	19 ± 4	12 ± 3	15 ± 2	24 ± 6	20 ± 5	22 ± 5	16 ± 3	20 ± 6	18 ± 3	16 ± 4	20 ± 5	19 ± 7
Elevated-plus maze												
Time (s) spent in central platform	100 ± 27	75 ± 26	94 ± 39	105 ± 35	101 ± 27	96 ± 39	74 ± 20	81 ± 14	74 ± 11	64 ± 18	81 ± 5	76 ± 10
Time (s) spent in closed arms	101 ± 36	99 ± 26	130 ± 42	155 ± 46	149 ± 57	126 ± 39	179 ± 8 **	151 ± 8 **	105 ± 39 ** +	66 ± 32 bb	118 ± 10 **	119 ± 14 **
N° of head-dippings	41 ± 9	43 ± 10	27 ± 12	21 ± 8	21 ± 9	29 ± 10	33 ± 10	44 ± 6	49 ± 8	55 ± 10 b	54 ± 10	57 ± 8
N° of rearings	12 ± 3	33 ± 8 *	43 ± 14 *	52 ± 15 aa	50 ± 19	42 ± 13	12 ± 4	13 ± 6	9 ± 6	20 ± 6 b	13 ± 2	14 ± 4

Each value represents the mean ± standard deviation of 6–15 values corresponding to the same number of animals. *: *p* < 0.05, **: *p* < 0.01, ***: *p* ≤ 0.001 with respect to the values obtained in their corresponding control group. +: *p* < 0.05, ++: *p* < 0.01 with respect to the values obtained in the corresponding NSC group. After a statistical analysis comparing the means with the Mann–Whitney U test for independent samples and the Chi-square test for categorical parameters, significant differences were observed in the indicated experimental groups. a: *p* < 0.05, aa; *p* < 0.01 with respect to old mice C. b: *p* < 0.05, bb: *p* < 0.01 with respect to PAM C. C: control group (old mice/PAM: n = 6; adult mice/E-NPAM: n = 7). NSC: non-skin-to-skin contact (old mice/PAM: n = 6; adult mice/E-NPAM: n = 15). SC: skin-to-skin contact (old mice/PAM: n = 6; adult mice/E-NPAM: n = 15).

**Table 2 ijms-24-04680-t002:** Immune parameters in peritoneal leukocytes of old mice/PAM and adult mice/E-NPAM after cohabiting for 15 min/day for 2 months.

Immune Parameters	Old Mice			Adult Mice			PAM			E-NPAM		
	C	NSC	SC	C	NSC	SC	C	NSC	SC	C	NSC	SC
Chemotaxis index of macrophages	320 ± 52	566 ± 281	445 ± 178	585 ± 163 aa	566 ± 164	625 ± 200	165 ± 5	189 ± 94	401 ± 45	1069 ± 300 bb	534 ± 260	1051 ± 529
Chemotaxis index of lymphocytes	207 ± 112	542 ± 244 *	735 ± 190 *+	710 ± 144 aa	619 ± 215	611 ± 291	306 ± 11	300 ± 17	268 ± 57	1141 ± 162 bb	295 ± 207 *	663 ± 249 *
Phagocytic efficacy (%)	31 ± 10	57 ± 17 *	51 ± 6 *	58 ± 11 a	54 ± 5	69 ± 7	184 ± 15	442 ± 60	322 ± 74	445 ± 209 b	371 ± 144	638 ± 252
Phagocytic index	93 ± 24	180 ± 68 *	146 ± 72 *	505 ± 185 aa	747 ± 93	695 ± 83	309 ± 19	408 ± 24 **	356 ± 31 **	378 ± 43 b	317 ± 64	412 ± 18 *+
LPS-stimulated lymphoproliferation (%)	90 ± 10	92 ± 7	100 ± 12	150 ± 15 a	143 ± 13	155 ± 12	84 ± 38	92 ± 24	94 ± 20	219 ± 62 b	139 ± 61	174 ± 42

Each value represents the mean ± standard deviation of 6–15 values corresponding to the same number of animals. *: *p* < 0.05, **: *p* < 0.01 with respect to the values obtained in their corresponding control group. +: *p* < 0.05, with respect to the values obtained in the corresponding NSC group. a: *p* < 0.05, aa; *p* < 0.01 with respect to old mice C. b: *p* < 0.05, bb: *p* < 0.01 with respect to PAM C. After a statistical analysis comparing the means with the Student’s t-test for independent samples, significant differences were observed in the indicated experimental groups. a: *p* < 0.05, aa; *p* < 0.01 with respect to old mice C. b: *p* < 0.05, bb: *p* < 0.01 with respect to PAM C. C: control group (old mice/PAM: n = 6; adult mice/E-NPAM: n = 7). NSC: non-skin-to-skin contact (old mice/PAM: n = 6; adult mice/E-NPAM: n = 15). SC: skin-to-skin contact (old mice/PAM: n = 6; adult mice/E-NPAM: n = 15). LPS: lipopolysaccharide.

**Table 3 ijms-24-04680-t003:** Redox parameters in peritoneal leukocytes of old mice/PAM and adult mice/E-NPAM after cohabiting for 15 min/day for 2 months.

Redox Parameters	Old Mice			Adult Mice			PAM			E-NPAM		
	C	NSC	SC	C	NSC	SC	C	NSC	SC	C	NSC	SC
GR activity (mU/mg protein)	12 ± 4	13 ± 2	15 ± 7	8 ± 1	10 ± 5	46 ± 5 bb	2.6 ± 1	1.3 ± 0.8	8 ± 2	1.6 ± 0.5	1.5 ± 1	6 ± 3
GPx activity (mU/mg protein)	5 ± 3	6 ± 2	20 ± 5 **++	40 ± 10 aa	35 ± 12	41 ± 9	4 ± 3	3.6 ± 2.8	5 ± 3	0.3 ± 0.3	0.5 ± 0.2	0.3 ± 0.1
GSH concentration (nmol/mg protein)	0.13 ± 0.03	0.09 ± 0.04	0.33 ± 0.1 *+	0.36 ± 0.2 a	0.26 ± 0.04	0.33 ± 0.16	0.14 ± 0.1	0.2 ± 0.05	0.24 ± 0.08	0.3 ± 0.2 b	0.28 ± 0.2	0.22 ± 0.1

Each value represents the mean ± standard deviation of 6–15 values corresponding to the same number of animals. *: *p* < 0.05, **: *p* < 0.01 with respect to the values obtained in their corresponding control group. +: *p* < 0.05, ++: *p* < 0.01 with respect to the values obtained in the corresponding NSC group. a: *p* < 0.05, aa; *p* < 0.01 with respect to old mice C. b: *p* < 0.05, bb: *p* < 0.01 with respect to PAM C. After a statistical analysis comparing the means with the Student’s t-test for independent samples, significant differences were observed in the indicated experimental groups. a: *p* < 0.05, aa; *p* < 0.01 with respect to old mice C. b: *p* < 0.05, bb: *p* < 0.01 with respect to PAM C. C: control group (old mice/PAM: n = 6; adult mice/E-NPAM: n = 7). NSC: non-skin-to-skin contact (old mice/PAM: n = 6; adult mice/E-NPAM: n = 15). SC: skin-to-skin contact (old mice/PAM: n = 6; adult mice/E-NPAM: n = 15). GR: glutathione reductase. GPx: glutathione peroxidase. GSH: reduced glutathione.

**Table 4 ijms-24-04680-t004:** Longevity results.

	Old Mice			Adult Mice			PAM			E-NPAM		
	C	NSC	SC	C	NSC	SC	C	NSC	SC	C	NSC	SC
Mean longevity (weeks)	87 ± 11	116 ± 11	96 ± 13	90 ± 5	108 ± 1	100 ± 9	73 ± 12	74 ± 17	97 ± 15 *+	110 ± 16 b	93 ± 19	103 ± 19
Maximum longevity (weeks)	112	115	155 *+	118	125	140 *	92	94	113 *+	129 b	125	122

Each value represents the mean ± standard deviation of 6–15 values corresponding to the same number of animals. After a statistical analysis comparing the means with the Student’s t-test for independent samples, significant differences were observed in the indicated experimental groups. *: *p* < 0.05 with respect to the values obtained in the corresponding control group (C). +: *p* < 0.05 with respect to the values obtained in the corresponding NSC group. b: *p* < 0.05 with respect to PAM C. C: control group (old mice/PAM: n = 6; adult mice/E-NPAM: n = 7). NSC: non-skin-to-skin contact (old mice/PAM: n = 6; adult mice/E-NPAM: n = 15). SC: skin-to-skin contact (old mice/PAM: n = 6; adult mice/E-NPAM: n = 15).

## Data Availability

Data will be provided if required.

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
