# Peer review of "Skin-to-Skin Contact: Crucial for Improving Behavior, Immunity, and Redox State after Short Cohabitation of Chronologically Old Mice and Prematurely Aging Mice with Adult Mice"

_ijms, 2023, doi:10.3390/ijms24054680_

Round 1

Reviewer 1 Report

The experiments are well designed and the results are sound. It's a well written manuscript, well organized in a comprehensive way.

Specific comments:

Results: Please, introduce also the number of animals used in each case in the legends of Figures and Tables

Methods: ¿How did you decide the time of cohabitation of mice?  Have you prove to increase the time of skin contact per week along the period of time used in the study?

Author Response

Reviewer #1: The experiments are well design and the results are sound. It’s a well written manuscript, well organized in a comprehensive way.

Specific comments:

Results: Please, introduce also the number of animals used in each case in the legends of Figures and Tables.

We thank the Reviewer for the suggestion. We have added the number of animals used in each case in the legends of the figures and tables in the revised version of the manuscript

Methods: How did you decide the time of cohabitation of mice? Have you proved to increase the time of skin contact per week along the period of time used in the study?

We thank these questions regarding this aspect. We would like to comment that this work is a continuation of other previous studies of our research group (Garrido et al., 2018,2019, 2020; Díaz-Del Cerro et al., 2022a, 2022b). Garrido et al (2018, 2019,2020) studied the effects of a cohabitation continuous for 2 months between old/ prematurely aging mice (PAM) and adult/ exceptional non-prematurely aging mice (E-NPAM). Díaz-Del Cerro et al. (2022a, 2022b) studied the effect of a cohabitation between old/ PAM and adult/E-NPAM for 15 min/day for 2 months. This change in the time of cohabitation is due to two reasons. First, Garrido et al (2018,2019,2020) showed that a continuous cohabitation between old/PAM and adult/E-NPAM for 2 months, had beneficial effects on the homeostatic systems of old and PAM, which was reflected in greater longevity, but adults and E-NPAM presented some deterioration in some of its parameters analyzed. Thus, in order to avoid this deleterious effect on adults and E-NPAM, we decided to shorten the time of social interaction to 15 min/day. In fact, this time the social interaction between chronologically old/PAM and adult mice/E-NPAM continued to exert a positive effect on the former without causing deterioration in adults (Diaz Del Cerro et al., 2022a, 2022b). Thus, we used this experimental design to study the effect of physical contact. Another reason for shortening social interaction time to 15min/day was to try to better extrapolate this strategy to humans, in which there is not frequently 24 hours of interaction a day between adults and the elderly. That interaction usually lasts a few hours. So, considering that the mean life span of ICR/CD1 female mice in our Animal House is 70 weeks and comparing it with the average life expectancy in humans, 1 week in mice would be equivalent to 1 year in humans. Therefore, 15 min would mean 8 hours of cohabitation in humans approximately.

Garrido A, Cruces J, Ceprián N, De la Fuente M. Improvements in Behavior and Immune Function and Increased Life Span of Old Mice Cohabiting With Adult Animals. J Gerontol A Biol Sci Med Sci. 2018;73(7):873-881. doi:10.1093/gerona/gly043

Garrido A, Cruces J, Ceprián N, Corpas I, Tresguerres JA, De la Fuente M. Social environment improves immune function and redox state in several organs from prematurely aging female mice and increases their lifespan. Biogerontology. 2019; 20(1), 49–69. doi: 10.1007/s10522-018-9774-4

Garrido A, Cruces J, Ceprián N, Díaz-Del Cerro E, Félix J, De la Fuente M. The ratio of prematurely aging to non-prematurely aging mice cohabiting, conditions their behavior, immunity and lifespan. J Neuroimmunol. 2020 Jun 15; 343:577240. doi: 10.1016/j.jneuroim.2020.577240.

Díaz-Del Cerro E, Ceprián N, Félix J, De la Fuente M. A short social interaction between adult and old mice improves the homeostatic systems and increases healthy longevity. Exp Gerontol. 2022 Feb; 158:111653. doi: 10.1016/j.exger.2021.111653.

Díaz-Del Cerro E, Félix J, De la Fuente M. Prematurely aging female mice improve their behavioural response, immunity, redox state, and lifespan after a short social interaction with non-prematurely aging mice. Biogerontology. 2022 Jun;23(3):307-324. doi: 10.1007/s10522-022-09968-9.

Reviewer 2 Report

The manuscript seeks to determine whether aging and premature aging can be inhibited by physical contact of old/prematurely aging mice with adult/exceptional non-prematurely aging mice.  The authors included control mice that were group housed as well as “NSC” mice that were cohabitated with adult/exceptional non-prematurely aging mice in cages with a barrier so that to direct contact was possible. The “skin contact” (SC) mice were cohabitated in a similar manner, but without the barrier to allow for physical contact. The authors evaluate the effect of this physical contact on certain behavioral tests, cellular immune functions, and the redox balance in immune cells isolated from the peritoneal cavity.  Animals allowed physical contact with either adult or exceptional non-prematurely aging mice performed more similarly to adults on most behavioral tests.  Immune cells isolated from aged/prematurely aging mice allowed physical contact with adults yielded functional and oxidative profiles more similar to adult mice than aging/ prematurely aging mice.  The authors examined each characteristic in multiple ways.

Overall, the study was well-designed and thoroughly presented.  Yet, information regarding how much physical contact each of the old/prematurely aging mice engaged in with the adult/exceptional non-prematurely aging mice during the 15-minute daily co-housing sessions is missing from the manuscript.  It is possible that the introduction of novelty (by sharing space with an adult/ exceptional non-prematurely aging mice), not physical contact, is responsible for the results obtained in the SC group.  An additional helpful control would be to include a 4th treatment group wherein old/ prematurely aging mice (that are not cage mates) share space with the SC group.  This would isolate novelty as its own treatment variable. Further, documenting the amount of physical contact engaged in by the old/prematurely aging mice in the SC group would strengthen the manuscript.  It is concerning that the treatment group is given the label “skin contact”, yet there is no data to support the idea that the mice engaged in physical contact.  Further, the label “skin contact” seems more appropriate for animals without fur rather than mice.

The cited references include an excessive number of self-citations (18 of the 33 citations include the senior author).  The manuscript would be strengthened by the inclusion of additional viewpoints, especially with respect to aging of the immune system and role of oxidization on this process.

The results section (lines 79-84) includes sentences that are a bit unclear.  Please separate these into two distinct sentences that clearly distinguish the data from old mice with SC from the PAM mice with SC. 

The authors present a large amount of data in the results.  Readers would be better able to follow the conclusions if the text of the results section utilized more detailed topic sentences and phrases that contextualized the results for old mice compared to the results for PAM mice.

It is difficult to read Table 1.  Values do not appear to line up correctly with the headings on the left.  It may be helpful to separate the behavioral, immunologic, and oxidative data into three separate tables.

Please rephrase the sentence on line 130 to “No difference” instead of “Any difference”.

Figure and table legends should include the type of statistical test performed to obtain the p values reported in that particular figure.

Please define abbreviations (GR, GPx, GSH, GSSG, etc.)

Author Response

Reviewer #2: The manuscript seeks to determine whether aging and premature aging can be inhibited by physical contact of old/prematurely aging mice. The authors included control mice that were group housed as well as “NSC” mice that were cohabited with adult/exceptional non-prematurely aging mice in cages with a barrier so that to direct contact was impossible. The “skin contact” (SC) mice were cohabited in a similar manner, but without the barrier to allow for physical contact. The authors evaluate the effect of this physical contact on certain behavioral tests, cellular immune functions, and the redox balance in immune cells isolated from the peritoneal cavity. Animals allowed physical contact with either adult o exceptional no-prematurely aging mice performed more similarly to adults on most behavioral tests. Immune cells isolated from aged/prematurely aging mice allowed physical contact with adults yielded functional and oxidative profiles more similar to adult mice than aging/prematurely aging mice. The authors examined each characteristic in multiple ways.

Overall, the study was well-design and thoroughly presented. Yet, information regarding how much physical contact each of the old/prematurely aging mice engaged in with the adult/exceptional non-prematurely aging mice during 15-minute daily co-housing sessions is missing from the manuscript. It is possible that the introductions of novelty (by sharing space with adult/exceptional non-prematurely aging mice), not physical contact, is responsible for the results obtained in the SC group. An additional helpful control would be to include a 4th treatment group wherein old/prematurely aging mice (that are not cage mates) share space with the SC group. This would isolate novelty as its own treatment variable. Further, documenting the amount of physical contact engaged in by the old/prematurely aging mice in the SC group would strengthen the manuscript. It is concerning that the treatment group is given the label “skin contact”, yet there is no data to support the idea that the mice engaged in physical contact. Further, the label “skin contact” seems more appropriate for animals without fur rather than mice.

We understand the Reviewer's doubts regarding this aspect. This work is the continuation of two previous studies (Díaz-Del Cerro et al., 2022a, 2022b). In these experiments, a group of old mice or PAM was put to interact for 15min/day for 2 months with another group of old mice or PAM, respectively. With this experimental group, we made sure that the results obtained by cohabiting 15min/day for two months with adult mice or E-NPAM were due to social interaction with individuals with a lower degree of aging and not to experimental management or by the introduction of these mice in a new environment. Indeed, in these two studies, we observed that the improvements in the behavioral, immune, oxidative, and inflammatory levels were due to social interaction with non-aged animals. Given these results, we decided not to introduce this experimental group in the present study and thus be able to use a smaller number of mice in order to satisfy the requirement in animal experimentation. Moreover, the mice in the experimental NSC group of the present study were experimentally manipulated in the same way as the mice in the SC group, and they were also exposed to a novel environment. However, there are almost no differences between C and NSC groups in the parameters studied. But the SC group shows several improvements in behavioral, immune response, and oxidative state with respect to groups C and NSC. Therefore, it can be deduced that the positive effects of the type of social interaction studied in this article are due, at least in part, to physical contact.

Despite all the above, we do not have data on the number of times there was physical contact between individuals in the cages during the time of the strategy. Therefore, we have added this aspect as a limitation of the study in the discussion section.

With respect to the comment of the Reviewer that there is no data to support the idea that the mice engaged in physical contact and thus the label “skin contact” is inappropriate, especially in animals with fur such as mice, we do not agree.  In our opinion, the term “skin contact” in the case of animals such as mice would be correct since the skin is considered a neuroimmunoendocrine organ thanks to all the components that form it, including the fur (Roosterman et al., 2006). Moreover, in the case of animals, the fur is associated with different nerve endings that make it a fundamental element for communication with different stimuli, as well as with other individuals (Mc Glone et al., 2014). Thus, although other terminology could be used in the case of these animals, such as “hair contact”, we opted for the term “skin contact” for including not only experimental animals in the message, but also humans. We want to emphasize that regardless of the species, skin contact is essential for the positive effect of social interaction to maintain the health.

Roosterman D, Goerge T, Schneider SW, Bunnett NW, Steinhoff M. Neuronal control of skin function: the skin as a neuroimmunoendocrine organ. Physiol Rev. 2006;86(4):1309-1379. doi:10.1152/physrev.00026.2005

McGlone F, Wessberg J, Olausson H. Discriminative and affective touch: sensing and feeling. Neuron. 2014;82(4):737-755. doi:10.1016/j.neuron.2014.05.001

The cited references include an excessive number of self-citations (18 of the 33 citations include the senior author). The manuscript would be strengthened by the inclusion of additional viewpoints, especially with respect of aging of the immune system and role of oxidization on this process.

We understand the Reviewer's commentary, so we have introduced cites from other authors.

The results section (lines 79-84) includes sentences that are a bit unclear. Please separate these into two distinct sentences that clearly distinguish the data from old mice with SC from the PAM mice with SC.

We agree with the Reviewer, so we have changed this part of the result section to: “However, old mice that skin-to-skin contacted adult mice improved these abilities since they took less time to complete the wood-rod test (Fig 1.A.1, p<0.05), and the percentage of animals with optimal traction was higher (Fig 1.B.1, p<0.05) than in the control groups (C and NSC groups). Similar effect can be observed in those PAM that skin-to-skin contacted E-NPAM (Fig 1.A.2 and Fig 1.B.2, p<0.05).

The authors present a large amount of data in the results. Readers would be better able to follow the conclusions if the text of the results section utilized more detailed topic sentences and phrases that contextualized the results for old mice compared to the results for PAM mice.

We thank the Reviewer for the suggestion. In order to clarify the results, we have added subtitles.

It is difficult to read Table 1. Values do not appear to line up correctly with the headings on the left. It may be helpful to separate the behavioral, immunologic, and oxidative data into three separate tables.

We thank the suggestion for the Reviewer, and we agree that it is better to separate the data into three tables for a better understanding.

Please rephrase the sentence on line 130 to “No difference” instead of “Any difference”

We thank the Reviewer for the suggestion. We have changed it in the revised version of the manuscript

Figure and table legends should include the type of statistical test performed to obtain the p-value reported in that particular figure.

We thank the Reviewer for the suggestion. We have added the type of statistical test performed in the legends of the figures and tables in the revised version of the manuscript

Please define abbreviations (GR, GPx, GSH, GSSG, etc)

We thank the reviewer for noticing this omission. We have added the definition of abbreviations in the legends of the figures and tables.

Round 2

Reviewer 2 Report

The authors have pointed out that the present study is a continuation of work published in references 14-18 and explain how this study differs slightly from previous studies.  The authors have also included a paragraph at the end of the manuscript pointing out that the experimental design did not including measuring the degree of physical contact between the cohabitated sets of mice. 

However, much of the work reported in the current manuscript overlaps with the work presented in references 14-18, with the exception that the current manuscript seeks to determine whether skin to skin contact is the necessary element for the improvements in behavioral test performance, immunologic parameters, and oxidative states of aged or PAM mice exposed for 15 min/day for 2 months to adult or ENPAM mice.  In my view, the authors have not definitively demonstrated that skin to skin contact even occurs in their study, yet this is essential to the conclusion.  The authors previous work does discount the original concern that the true variable may be novelty, but does not eliminate other possibilities.  For example, the microbiome of mice changes as they age and it is certainly possible that the critical element of the 15 min exposures over the two months was exposure to the adult or ENPAM mice's feces/dirty bedding. 

The authors indicated that they have broadened their citations to include perspectives beyond the contributing authors contributions, but no new citations appear to be included.

Similarly, the authors have indicated the sentence on line 187 was changed to “No difference” from “Any difference”, but this change does not appear on my version of the manuscript.

The authors included mention of the statistical tests used in each figure legend (student's t test), but the methods section does not include this test.  Further, a student's t test does not appear to be the appropriate statistical test to use, considering each experiment includes more than two experimental groups.

Author Response

International Journal of Molecular Science

Ms. Ziva Wang

16th February 2023

Dear Ms. Ziva Wang,

We are enclosing a second revised version of the manuscript " Skin-to-skin contact: crucial for improving behavior, immunity, and redox state after short cohabitation of chronological old mice and prematurely aging mice with adult mice" (ID: ijms-2153641).

We thank the Reviewer and the Editors of the International Journal of Molecular Science for their encouraging comments, all of which have been considered to correct and improve the manuscript.

Replies to comments have been structured in the same order as were commented by the Reviewer, and all changes have been highlighted in yellow in the revised manuscript version.

Reviewer #1: The authors have pointed out that the present study is a continuation of work published in references 14-18 and explain how this study differs slightly from previous studies.  The authors have also included a paragraph at the end of the manuscript pointing out that the experimental design did not include measuring the degree of physical contact between the cohabitated sets of mice.

However, much of the work reported in the current manuscript overlaps with the work presented in references 14-18, with the exception that the current manuscript seeks to determine whether skin-to-skin contact is the necessary element for the improvements in behavioral test performance, immunologic parameters, and oxidative states of aged or PAM mice exposed for 15 min/day for 2 months to adult or ENPAM mice.  In my view, the authors have not definitively demonstrated that skin-to-skin contact occurs in their study, yet this is essential to the conclusion.  The author’s previous work does discount the original concern that the true variable may be a novelty but does not eliminate other possibilities.  For example, the microbiome of mice changes as they age and it is certainly possible that the critical element of the 15 min exposures over the two months was exposure to the adult or ENPAM mice's feces/dirty bedding.

We thank the Reviewer´s comments, but we do not agree that this study has not shown that skin-to-skin contact occurs. Although we did not record and count the number of physical contacts during the 15 minutes of interaction, we observed that this type of contact did indeed occur among the mice. However, we agree that it is a limitation of the study and, therefore, we add it to the discussion. With respect to the fact that there may be other possible mechanisms that mediate the effects observed in this type of coexistence, indeed, it is. We do not propose in our article that skin-skin contact is the only mechanism that causes the positive effects of this type of social interaction, but we do observe that it is necessary for them to occur. Other mechanisms, along with physical contact, can have a synergistic effect, such as smell or hearing. We do not think that it can be due to the exchange of microbiome because we observe that in the 15 minutes of interaction, the mice barely defecate, and we do not observe coprophagous behaviors either. However, we have added the following paragraph in the discussion to clarify this aspect of the experiment: “Another limitation presented by the study is that other mechanisms that are also involved in the beneficial effects obtained in this type of social interaction may be having a summative effect with physical contact. In fact, some of these mechanisms may be visual, olfactory, and auditory perception as they change with advancing age or health status [30-33]. Another possible mechanism could be the ingestion of fecal balls, which would lead to the alteration of the microbiota [34]. However, we see this mechanism as being more possible if the cohabitation was continuous, but with interaction rates of 15 minutes a day, the mice hardly defecated, and we did not observe coprophagous behaviors.”

The authors indicated that they have broadened their citations to include perspectives beyond the contributing authors contributions, but no new citations appear to be included.

We are sorry that in your version of the manuscript, the included citations did not appear, we hope that in the new version, you can observe it without a problem. In fact, we have added the following references:

[3] Liguori I, Russo G, Curcio F, et al. Oxidative stress, aging, and diseases. Clin Interv Aging. 2018;13:757-772. Published 2018 Apr 26. doi:10.2147/CIA.S158513

[4] Spencer RL, Hutchison KE. Alcohol, aging, and the stress response. Alcohol Res Health. 1999;23(4):272-83. PMID: 10890824; PMCID: PMC6760387.

[5] Aschbacher K, O'Donovan A, Wolkowitz OM, Dhabhar FS, Su Y, Epel E. Good stress, bad stress and oxidative stress: insights from anticipatory cortisol reactivity. Psychoneuroendocrinology. 2013;38(9):1698-1708. doi:10.1016/j.psyneuen.2013.02.004

[15] Deak T, Kudinova A, Lovelock DF, Gibb BE, Hennessy MB. A multispecies approach for understanding neuroimmune mechanisms of stress. Dialogues Clin Neurosci. 2017;19(1):37-53. doi:10.31887/DCNS.2017.19.1/tdeak

[30] Salchner P., Lubec G., Singerwald N. Decreased social interaction in aged rats may not reflect changes in anxiety-related behaviour. Behav. Brain research. 2004. (151): 1-8. DOI:  10.1016/j.bbr.2003.07.002

[31] Finkel J.C., Besch V.G., Hergen A., Kakarena J., Pohida T. Effects of aging on current  vocalization threshold in mice measured by a novel nociception assay. Anesthesiology. 2006. (105): 360-9.

[32] Osada K., Yamazaki K., Curran M., Bard J., Smith B.P.C., Beauchamp G.K. The scent of age. Proc. R. Soc. Lond. B. 2003. (270): 929-33

[33] Brudzynsky S.M. Ethotransmission: communication of emotional states through ultrasonic vocalization in rats. Curr. Op. Neurobiol. 2013. (3): 310-7 DOI: 10.1016/j.conb.2013.01.014

[34] Foster J.A., Rinaman L., Cryan J.F. Stress and the gut-brain axis: Regulation by the  microbiome. Neurobiol. Stress. 2017. (7): 124-36. DOI: 10.1016/j.ynstr.2017.03.001

Similarly, the authors have indicated the sentence on line 187 was changed to “No difference” from “Any difference”, but this change does not appear on my version of the manuscript.

We are sorry that you did not observe the changes in your version of the manuscript. We hope that in this new revised version of the article you can see that we have made the changes suggested from the beginning.

The authors included mention of the statistical tests used in each figure legend (student's t test), but the methods section does not include this test.  Further, a student's t test does not appear to be the appropriate statistical test to use, considering each experiment includes more than two experimental groups.

Again, we regret that you did not correctly observe the changes made in the previous version of the manuscript. In this new version, you can see in the Material and Methods section that this statistical analysis is included: “…Subsequently, a comparison of the means was carried out via a non-parametric analysis (Mann–Whitney's U test) and a parametric analysis (Student’s t test) of independent variables to identify the differences between the study groups. In addition, the categorical variables were analyzed using a contingency test (Chi-square) ...”

Although the experiment includes more than two experimental groups, we decided on this type of statistical analysis by sample size. We consider that the n of the experimental groups is not enough to perform another type of statistical test such as ANOVA.

Finally, the manuscript has been English edited by MDPI, following the suggestions of the Reviewer and the Editor. We hope that this revised version of the manuscript is suitable for publication in the International Journal of Molecular Science.

We look forward to receiving your feedback.

Yours sincerely,

Prof. Dr. Mónica De la Fuente

Department of Genetics, Physiology, and Microbiology

Faculty of Biology, Complutense University of Madrid

Email address: [email protected]

Jose Antonio Novais Street 2, 13th floor 28040 Madrid, Spain

Telephone number: +34 91 3944989.

Round 3

Reviewer 2 Report

The additional paragraph the authors added to the discussion addresses the main concern this reviewer had with the manuscript.  I believe the new paragraph contextualizes the findings and softens the conclusions appropriately.

I was also able to view the other changes that I did not observe in the previous version.